# Mediterranean Diet to Prevent the Development of Colon Diseases: A Meta-Analysis of Gut Microbiota Studies

**DOI:** 10.3390/nu13072234

**Published:** 2021-06-29

**Authors:** Oscar Illescas, Miriam Rodríguez-Sosa, Manuela Gariboldi

**Affiliations:** 1Genetic Epidemiology and Pharmacogenomics Unit, Department of Research, Fondazione IRCCS Istituto Nazionale dei Tumori (INT), 20133 Milan, Italy; oscarillescas.pomposo@istitutotumori.mi.it; 2Unidad de Biomedicina, Facultad de Estudios Superiores Iztacala, Universidad Nacional Autónoma de México (UNAM), Tlalnepantla C.P. 54090, MEX, Mexico; rodmir@unam.mx

**Keywords:** microbiota, 16S, meta-analysis, Mediterranean diet, inflammation, adenoma, colorectal cancer

## Abstract

Gut microbiota dysbiosis is a common feature in colorectal cancer (CRC) and inflammatory bowel diseases (IBD). Adoption of the Mediterranean diet (MD) has been proposed as a therapeutic approach for the prevention of multiple diseases, and one of its mechanisms of action is the modulation of the microbiota. We aimed to determine whether MD can be used as a preventive measure against cancer and inflammation-related diseases of the gut, based on its capacity to modulate the local microbiota. A joint meta-analysis of publicly available 16S data derived from subjects following MD or other diets and from patients with CRC, IBD, or other gut-related diseases was conducted. We observed that the microbiota associated with MD was enriched in bacteria that promote an anti-inflammatory environment but low in taxa with pro-inflammatory properties capable of altering intestinal barrier functions. We found an opposite trend in patients with intestinal diseases, including cancer. Some of these differences were maintained even when MD was compared to healthy controls without a defined diet. Our findings highlight the unique effects of MD on the gut microbiota and suggest that integrating MD principles into a person’s lifestyle may serve as a preventive method against cancer and other gut-related diseases.

## 1. Introduction

The increased incidence and mortality of many cancers observed in the last few decades has been in part attributed to the modern and heavily industrialized lifestyle [1,2], characterized by environmental pollution, stress, sedentarism, and a diet dominated by pre-processed foods rich in fat, salt, meat, refined flour and sugar, and little to no fresh vegetables, fruits and nuts, such as the so-called Western-type or Westernized diet (WD) [3,4,5].

Lifestyle-changing therapeutic application of healthier dietary patterns has been considered effective, non-invasive, and long-lasting [6,7,8], and several diets have been proposed for the treatment or prevention of different diseases. One of the most widely used is the Mediterranean diet (MD). This term encompasses a series of dietary patterns used in countries of the Mediterranean coast, with shared common premises including high consumption of fresh vegetables, fruits, nuts, grains, legumes and olive oil, moderate to high consumption of fish, moderate consumption of dairy and wine, and low intake of meat and sweets [9]. The benefits of MD have recently been suggested on several pathologies, including cancer [10,11,12,13].

The modern Paleolithic diet (PD) is a different approach characterized by a high intake of vegetables, fruits, nuts, eggs, fish and meat, minimizing that of grains, cereals, legumes, dairy products, and processed foods [14]. Recently, it has sparked increasing interest due to its alleged beneficial effects on the prevention and treatment of diabetes, cancer, and cardiovascular diseases [15,16,17,18].

In terms of microbiota composition, both MD and PD reduce the consumption of simple sugars, animal fats, and processed foods compared to WD, and this alone is known to have a potentially beneficial impact on microbiota and health [19,20,21,22]. However, MD differs from PD by including carbohydrate sources such as grains and legumes, which PD excludes, and by reducing the intake of animal-based protein, which has also been shown to influence the microbiota composition [23,24,25,26].

It has been postulated that many of the effects of dietary interventions are driven by the microbiota. Dietary components can shape the composition of the gut microbiota and modulate its metabolism [27], which in turn will affect the production of beneficial metabolites such as short-chain fatty acids (SCFA) and potentially toxic compounds such as procarcinogen trimethylamine (TMA), or also modulate the metabolism and immune response at the systemic and local level [28,29,30,31,32]. Under normal conditions, the microbiota provides metabolic pathways that interact and complement the host metabolism, helping to maintain normal organ function and nutrition while limiting the activity of pathogens.

Contrary to this normal state, the term dysbiosis refers to an imbalance in the composition and metabolism of the microbiota that has been associated with pro-inflammatory conditions and therefore has been suggested to contribute to the pathogenesis of various diseases, ranging from psychiatric diseases to metabolic disorders and cancer [31,33,34]. This imbalance and associated inflammation is often attributed to WD [3,4], and it plays an important role in the pathogenesis of inflammatory bowel diseases (IBD) such as Crohn’s disease (CD) and ulcerative colitis (UC) as well as colorectal cancer (CRC) [35,36,37]. Importantly, in recent years, the incidence of all these diseases has increased in industrialized and newly industrialized countries [38,39].

Trying to define and describe the complex relations between diet, microbiota, and disease, several cross-sectional and interventional studies describing the microbiota associated with specific dietary patterns or diseases have been conducted. No attempt has yet been made to cross data from diet and disease studies to identify common or specific bacterial populations that could be proposed for preventing gut-related diseases.

We performed a joint meta-analysis of 16S RNA data from subjects on diets such as MD, PD, or WD, and patients with intestinal diseases related to inflammation or cancer. We aimed to determine whether the microbiota composition linked to any of the diets may justify a possible preventive or therapeutic use of the diet. This is by finding the bacteria differently represented in each diet or disease and evaluating their reported health benefits or pathogenic functions.

## 2. Materials and Methods

### 2.1. Data Acquisition and Inclusion Criteria

The 16S rRNA gene datasets included in the meta-analysis are publicly available and were identified through a literature search in NCBI PubMed [40] and in the Sequence Read Archive [41]. Search terms were “gut microbiota” or “16s” together with one of the following: “colitis”, “IBD”, “Crohn’s disease”, “colorectal cancer”, “colon cancer”, “colon adenoma”, “adenoma”, “colon polyp”, “Mediterranean diet”, “Western diet”, “Westernized diet”, “paleolithic diet”, “fodmap”, and “ketogenic diet”. The search was limited to articles published from January 2008 to July 2020. When the study did not include a public availability statement, the corresponding author was contacted to ask for access to the data, but we did not include any studies that required additional ethics committee approvals or authorizations for access (e.g., controlled dbGaP studies). Inclusion criteria for all included datasets were as follows:Cross-sectional studies or dietary interventions with MD, PD, or WD, or case-control studies of intestinal diseases such as colorectal cancer (CRC), colon adenoma (CA), colon polyposis (CP), ulcerative colitis (UC), and Crohn’s disease (CD). May include or not healthy subjects (HC), healthy familiars of IBD subjects (FC), or subjects at risk of developing CRC (RS).Available 16s rRNA gene sequences derived from human feces with associated quality scores and metadata.Samples obtained when subjects or patients were not under drug treatments.Data from at least 10 patients, subjects following a defined diet or healthy subjects with a minimum of 6000 reads each.

### 2.2. Study Groups

Subjects from the obtained databases were assigned to different groups, defined according to the subject’s dietary pattern and (or) diagnosed gastrointestinal disease. Either cross-sectional or interventional studies on MD, PD, and WD diets were included in the analysis. Diet composition and the main hallmarks of each diet were obtained directly from the included publications. WD featured a low content of fiber, fruits, and vegetables contrasted by a high consumption of refined carbohydrates and saturated fats [42]. PD was characterized by the consumption of vegetables, fruits, seeds, lean meat, eggs, and fish, the reduction of salt and refined sugars, and the exclusion of grains, pulses, and dairy products [43]. Adherence to MD was defined with validated scores [44,45].

Case-control cross-sectional studies and pre-treatment data from clinical trial reports on CRC, CA, CP, UC, and CD were included; guidelines for the diagnosis of all diseases are reported in the corresponding study. HC, FC, and RS subjects were obtained when available in any of the included studies, none of them followed any specific or controlled diet. HC and FC subjects were all healthy, RS subjects were defined as at risk of developing CRC after colonoscopy, had previous personal or familiar history of CRC, or presented obesity (BMI ≥ 30, Appendix A). None of the subjects in the HC, FC, or RS groups followed a defined diet. Additional studies [27,46,47,48] were included with the intention of increasing the number of control subjects.

### 2.3. Data Analysis and Statistics

Raw sequences were processed and analyzed in QIIME 1.9.1 [49]. Sequences were quality filtered with Trimmomatic [50] by truncating when the average quality in a four-base sliding window dropped below Q < 25. Surviving high-quality reads were sorted with 99% similarity into operational taxonomy units (OTUs) with uclust [51] and aligned against the Greengenes 13.8 database with 97% similarity using PyNAST [52,53]. Chimeric sequences were detected with ChimeraSlayer [54] and excluded from downstream analysis. This stringent processing and analysis resulted in several processed samples not meeting the initial inclusion criteria; therefore, samples with less than 6000 surviving reads and datasets with less than 10 subjects per group were dropped. OTUs with less than 10 reads were also discarded for further analyses.

All data were analyzed together and with the same parameters. Taxonomy assignment was performed with a widely used and trusted reference database covering all included variant regions of the 16s rRNA gene [53]. Sequences that failed to align to the database were not considered. Alpha diversity was assessed using Chao1 metric, total observed OTUs, and the Shannon diversity index. Significance of the test was determined with a non-parametric T-test, and *p*-values were corrected with the false discovery rate (FDR) method (Appendix A). Only comparisons presenting *p* < 0.001 were considered significant. Beta diversity was estimated using Unifrac distance metrics [55]. Jackknife-supported Principal Coordinates Analyses (PCoA) were performed with QIIME and visualized with Past 4.03 [56] (Appendix A). The significance of PCoA data separation was verified with a permutation test with pseudo-F ratios (function ADONIS) and an analysis of similarities (function ANOSIM) (Appendix A).

OTU representation was summarized at the phylum, family, and genus levels following the QIIME pipeline (Appendix A); only fully annotated OTUs were considered for analysis. Fold change was calculated in all instances using the abundance of group HC as a reference. The Wilcoxon rank sum test was used to assess differentially represented OTUs; *p*-values were corrected for FDR and considered statistically significant when *p* < 0.01 (Appendix A). All statistical analyses were performed in QIIME.

## 3. Results

A total of 168 clinical or observational studies on diet or gut diseases-associated microbiota, 42 of which included 16S rRNA analysis, were initially identified. Datasets corresponding to 26 studies were available; two of them were discarded, as samples were not sufficiently annotated. Another seven studies did not reach the minimum threshold of 10 samples with at least 6000 quality-checked reads and were discarded.

The remaining 17 studies were included in the meta-analysis. Included datasets comprised 1931 human fecal samples for a total of 157,425,716 reads; 80.9% of the samples (1563) and 95.7% of reads (150656411) survived the quality check and were used for the analyses. Samples were assigned to eleven different groups (Table 1 and Appendix A).

### 3.1. Alpha and Beta Diversity

Alpha diversity presented no significant differences (*p* < 0.001) between either diet and disease groups or respect to the HC controls (Appendix A).

Beta diversity components of groups HC, FC, and RS, observed in PCoA, presented no significant separation. Instead, MD distanced from WD, PD, HC, and RS groups. Likewise, CRC and CA formed separated clusters, which were partially segregated from RS and HC. UC and CD clustered together but remained separated from HC and FC. MD also segregated when confronted to CA, CRC, UC, and CD. In all cases, the separation was more evident with unweighted data, which only take into account the presence of OTUs, compared to weighted data, which consider both the presence and abundance of OTUs. (Figure 1, Appendix A). These distributions suggest that the microbiota composition of the subjects following MD is different from diseases and controls, and this effect is not shared by the other diets.

### 3.2. Microbiota Composition Analysis

Average proportional composition of all groups by phyla and genera can be found in Figure 2 and Table 2 and Table 3; only significant differences (*p* < 0.01) were discussed (Appendix A).

Comparison between groups showed that MD-induced microbiota is different from that of the other diets, the controls without a defined dietary pattern (HC), and the RS group (Table 2 and Table 3, Figure 2).

Out of the 12 most abundant phyla, nine were differently represented in MD compared to WD and four were differently represented in MD compared to PD (Appendix A). The high data variability of the PD group reduced the significance of its differences with other groups. Still, MD differed from both PD and WD by an increase in Verrucomicrobia (*p* = 0.0099 and *p* = 5.6 × 10^−6^ respectively), and from WD alone by a higher representation of Bacteroidetes (*p* = 7.9 × 10^−7^) and lower Firmicutes (*p* = 0.0002), Euryarchaeota (*p* = 0.0003) and Fusobacteria (*p* = 2.4 × 10^−10^), among other phyla. MD also presented nine differently represented phyla to RS and FC, and seven to HC, including a lower representation of Fusobacteria (*p* = 0.0027, *p* = 0.0051 and *p* = 0.0002 respectively) and higher of Verrucomicrobia and Actinobacteria (*p* < 0.0001 for all).

Among diseases, the CRC group distinguished from CA and CP by a higher presence of Proteobacteria, Fusobacteria, Euryarchaeota, and Verrucomicrobia (*p* < 0.0001 for all). The representation of these phyla was also higher in CRC than in HC (Proteobacteria *p* = 0.005, Fusobacteria *p* = 0.0051, Euryarchaeota *p* = 0.005 and Verrucomicrobia *p* = 0.0082), RS (*p* < 0.0001 for all). Chronic inflammation-related UC and CD did not present statistically significant differences between them at any taxonomic level (Appendix A); therefore, the two groups were merged for further comparison (IBD group). The IBD group presented seven differently represented phyla than HC among the 12 most abundant, but only three to FC, with a higher proportion of Firmicutes (*p* = 0.0002 and *p* = 2.5 × 10^−5^ respectively) and Actinobacteria (*p* = 0.002 and *p* = 0.0006) against both, and also higher Fusobacteria (*p* = 0.002) and lower Verrucomicrobia (*p* = 0.0002) compared to HC alone. The number of differentially represented phyla among the 12 most abundant in MD compared to CA and CRC was nine and ten respectively, with MD presenting higher Actinobacteria and lower Proteobacteria and Fusobacteria compared to both (*p* < 0.0001 for all). Differently, MD showed a higher proportion of Verrucomicrobia than CA (*p* = 3.5 × 10^−14^) but lower than CRC (*p* = 7.3 × 10^−22^). Likewise, compared to IBD, MD had nine differentially represented phyla, including a higher proportion of Actinobacteria (*p* = 0.0001) and Verrucomicrobia (*p* = 4.9 × 10^−5^), and lower of Proteobacteria (*p* = 0.0003) and Fusobacteria (*p* = 2.6 × 10^−12^).

Some of the phyla described showed progressive increases or decreases in their relative abundance from subjects in the healthy, at-risk, and cancer-related groups, placing MD furthest from CRC (Figure 3).

The differences observed in some phyla were also present at genus levels. Actinobacteria phylum is composed by several genera differently represented between groups, being the most abundant *Bifidobacterium*, *Collinsella*, *Adlercreutzia*, and a non-annotated OTU within family *Coriobacteriace* (Figure 4). *Adlercreutzia* was higher in MD with respect to HC, RS, CA, CRC, and IBD (*p* = 0.0011, *p* = 0.0024, *p* = 0.0003, *p* = 0.0002 and *p* = 0.007, respectively), while *Collinsella* was increased in IBD compared to HC, FC, and MD (*p* = 1.3 × 10^−6^, *p* = 0.0074 and *p* = 0.0086, respectively). Finally, *Bifidobacterium* was decreased in CRC compared to HC, RS, and MD (*p* = 0.0005, *p* = 2.5 × 10^−5^ and *p* = 1.5 × 10^−18^, respectively).

Analysis of genera also showed differences between groups that were not evident at the family level. The [*Ruminococcus*] genus from *Lachnospiraceae* was less represented in MD compared to HC (*p* = 3.6 × 10^−21^), FC (*p* = 1.5 × 10^−9^), and RS (*p* = 1.2 × 10^−7^), while it was increased in both IBD (*p* = 0.0097) and CRC (*p* = 4.4 × 10^−5^) against HC. *Veillonellaceae* is mainly constituted by the genera *Dialister*, *Veillonella,* and *Phascolarctobacterium*, which were the most abundant in MD, IBD, and CRC respectively (Figure 5).

Other phyla such as Verrucomicrobia, Euryarchaeota, and Fusobacteria were prevalently constituted by a single genus: *Akkermansia*, *Methanobrevibacter,* and *Fusobacterium,* respectively (Appendix A).

## 4. Discussion

The Mediterranean diet is widely accepted as part of a healthy lifestyle. Several studies, including those from our meta-analysis, show that the microbiota of MD subjects is enriched in bacteria with beneficial properties that help in maintaining gut barrier function and reducing inflammation. Instead, pathogenic bacteria with pro-inflammatory properties, which can impair the epithelial barrier function and produce toxic metabolites, are poorly represented. The gut microbiota of IBD, adenoma, and CRC shows an unbalanced bacterial composition that contributes to disease progression. We investigated if the bacterial population induced by MD may have preventive properties against these diseases.

Beta diversity analysis confirmed that microbiota associated to MD was different from that of the disease and control groups, and this was not observed with the other diets. Analysis of differently represented OTUs between groups found bacteria with a tendency to increase or decrease along an axis formed by MD–HC–RS–CA–CRC groups. In agreement with the hypothesis of microbiota components accompanying the formation of adenomas and the development of CRC, phyla with pro-inflammatory properties such as Proteobacteria and Euryarchaeota increased along the axis. A similar trend was also observed for Fusobacteria, which are known to promote colorectal carcinogenesis. In contrast, Actinobacteria, which include SCFA producers with anti-inflammatory properties, decreased (Figure 3).

The limited number of studies available on PD, WD, and CP resulted in the high variability and lack of statistical significance on these groups, and they did not consistently fit the axis. No statistical difference was found between PD and HC at the phylum level. Instead, WD presented a high variability and may be considered part of the axis in some cases, such as Fusobacteria, but not in others, such as Actinobacteria (Appendix A). CP was found many times at similar levels to HC and RS, for example, in Proteobacteria and Fusobacteria. The IBD group, made up of subjects obtained from three different studies, had characteristics that could be considered intermediate between HC and CRC, because in most cases, it presented abundance levels similar to the first, as in Actinobacteria, or to the latter, as in Fusobacteria.

The Proteobacteria phylum is a marker of gut dysbiosis [68], and one of the phyla increased along the axis. It has been associated with a high-fat diet and obesity in mice [69], with consumption of animal fat [70], and most importantly, with a colonic mucosa more easily penetrated by other bacteria [71]. This may explain the low abundance observed in MD, where animal fat consumption is greatly diminished, but not in PD, which includes several animal-derived products. Still, Proteobacteria is the most differently represented phyla between MD and PD (*p* = 8 × 10^−5^) and coincidently also between MD and HC (*p* = 2.6 × 10^−20^) (Appendix A). The decrease of Proteobacteria in PD compared to HC was not significant (*p* = 1). The increasing trend along the axis is found further down at the family level, driven by *Enterobacteriaceae*, whose enrichment in the gut is favored by inflammation [72]. This family has previously been found increased in patients with CRC and IBD [73,74] and is a proposed marker of epithelial dysfunction [75]. Other families within Proteobacteria are also decreased in MD compared to controls and diseases, including *Sphingomonadaceae* and *Helicobacteraceae*. *Sphingomonadaceae*’s most abundant genus, *Sphingomonas*, was found increased in IBD patients and is associated with CD recurrence [76,77], and it increased in colitis-associated CRC patients as well [78]. The *Helicobacteraceae* family includes the pathogen *Helicobacter pylori*, whose prevalence is increased in patients with gastric cancer [79]. This bacterium promotes its pathogenesis by inducing chronic inflammation, accumulation of mutations, and aberrant DNA methylation in the gastric mucosa through the expression of different virulence factors [80,81,82]. It is also found increased in CA and CRC patients [83], and its participation in CRC development has been suggested [84,85].

The most abundant Fusobacteria species, *Fusobacterium nucleatum*, is a pathogen increased in IBD and CRC [86,87,88]. *Fusobacterium* suppresses the immune cell response in the gut while also promoting a pro-inflammatory and tumorigenic environment. It produces the virulence factor FadA, which is capable of binding E-cadherin and activating beta-catenin [89], and Fap2, which can inhibit NK and T cell activity [90]. It also recruits several tumor-promoting cells, such as myeloid-derived suppressor cells and tumor-associated neutrophils and macrophages [91]. *F. nucleatum* promotes the destruction of the mucosal barrier by activating the endoplasmic reticulum stress pathway [92], and it has been associated with accelerated DNA methylation in the mucosa of UC patients [93]. The low abundance of *Fusobacterium* in MD may be associated with augmented dietary fiber and reduced fat consumption, as reported by studies on similar dietary patterns [94,95]. The reduction in abundance of a pathogen with such a significant role in cancer is indicative of the potential preventive function of the MD diet. Fusobacteria was also the most significant differently represented phylum between MD and WD (*p* = 2.4 × 10^−10^), and its greater representation in the latter when compared to HC may be indicative of a dietary adverse effect.

The only *archaea* phylum in the meta-analysis, Euryarchaeota, also increased along the MD–HC–RS–CA–CRC axis, except for RS (Figure 3). Euryarchaeota’s main component, *Methanobrevibacter smithii,* is the dominant *archaea* species in the human gut, and its abundance correlates with the activation of several pro-inflammatory pathways in multiple sclerosis [96]. It is increased in CA and CRC patients [97] and decreased in obese individuals [98], which may explain the reduction observed in RS, which is a group defined by obesity and other commonly related clinical markers. According to the literature, *Methanobrevibacter* presented a low abundance in MD subjects [99]. The presence of *Methanobrevibacter* has been associated with the consumption of dietary fiber [100], but dairy products, which are consumed with moderation in MD, are also possible sources of *M. smithii* [101]. A similar low representation was found for PD, although the decrease was non-significant compared to HC.

An abundance of Actinobacteria has been associated with the consumption of dietary fiber and production of SCFAs [102,103]. These anti-inflammatory compounds are a source of energy for epithelial cells and help maintain the stability of the gut barrier [28,104]. *Bifidobacterium* and *Adlercreutzia*, two commensals with anti-inflammatory properties that are used as probiotics [105,106], were the most abundant Actinobacteria genera in MD (Figure 4). The most common *Adlercreutzia* species is *A. equolifasciens*, which can produce the anti-inflammatory molecule equol [106,107]. Equol-producing microbiota is found in 59% of vegetarians compared to 25% in non-vegetarians [108]. The increase in vegetables consumption, together with the inclusion of *omega* 3-rich fat, which has also been related to an increase in *Adlercreutzia* [109], may explain its abundance in MD. We also found an increase of Actinobacteria in IBD subjects compared to FC and HC, confirming previous reports [110], but the increase was driven by the genera *Bifidobacterium* and *Collinsella*, and not by *Adlercreutzia* (Figure 4). Both *Collinsella* and *Adlercreutzia* are found in the Coriobacteriaceae family, but the bile-tolerant *Collinsella* has been linked to pro-inflammatory diseases, obesity, and low fiber consumption [111,112,113]. Another Actinobacteria genus increased in MD was *Slackia*, which was previously found increased after an intervention with a modified Mediterranean-ketogenic diet [114], and it includes species such as *S. equolifasciens* and *S. isoflavoniconvertens*, which are capable of producing equol [115,116] but also the opportunistic *S. exigua* [117]. Actinobacteria phylum was found to be increased in the WD group compared to HC and PD, and indeed, this was the most differently represented phylum between PD and WD, but the increase was not significant compared to MD. At the genus level, WD had the second highest abundance of *Collinsella*, higher than MD, HC, and even RS, and only below that of the IBD group.

Another phylum that decreased along the MD–HC–RS–CA–CRC axis is TM7, which is currently known as *candidatus* Saccharibacteria, as it seems to consume mainly sugars [118], but studies on its potential role and activity are scarce. They are obligate epibionts/parasites of other bacteria, and their increase has been reported in the context of IBD [119], although the first successfully cultured strain repressed the expression of inflammatory TNF-alpha induced by its host [120]. Unfortunately, the lack of information on this phylum does not offer a possible explanation for its apparent increase in MD subjects.

The low fat and high dietary fiber content typical of MD foster an environment that favors the growth of anti-inflammatory bacteria and hinders that of pro-inflammatory and pathogenic bacteria. This is also evident when MD is compared to HC and highlights its beneficial effect on gut health.

Other phyla that did not follow the axis are also modulated by diet components typical of MD, but differences are observed at lower taxonomic levels. This was the case for families and genera within Firmicutes, including *Ruminococcaceae*, which had increased in MD and PD compared to all the other groups (Table 3). Several species of butyrate-producing bacteria are found within this family [121], and its abundance in MD and PD is most likely linked to the high consumption of dietary fiber, from which SCFAs are derived.

Two other Firmicutes families with both pro-inflammatory and anti-inflammatory components, *Lachnospiraceae* and *Veillonellaceae*, had differences at the genus level. *Lachnospiraceae* was lower in the MD group than in IBD and CRC (Figure 5). Regarding the genera included in this family, [*Ruminoccocus*], it was increased in IBD and CRC. These mucolytic bacteria have been found increased in IBD subjects, and one of its species, *R. gnavus,* synthesizes a pro-inflammatory polysaccharide [122,123,124]. Instead, butyrate producers with local anti-inflammatory effect, belonging to *Roseburia*, are depleted in both IBD and CRC [125,126,127]. MD and also PD were enriched in genera with anti-inflammatory properties, including *Coprococcus,* which can produce indole-propionic acid, which is a tryptophan-derived metabolite with antimycobacterial, antioxidant, and anti-inflammatory activities [128,129]. Furthermore, *Dorea* was increased in MD, while PD was enriched *in Lachnospira*. Both genera include some SCFA-producing pectin fermenters and are associated with vegetarian and vegan diets [130,131,132], but they have also been reported to increase in obese subjects [133,134]. *Blautia* is a butyrate producer inversely related to intestinal inflammation [135] but was nevertheless found increased in IBD.

The family *Veillonellaceae* was increased in MD, CRC, and IBD compared to HC. The increase in MD was mainly driven by the genus *Dialister*, in CRC by *Phascolarctobacterium*, and in IBD by *Veillonella* (Figure 4). Bacteria belonging to these genera are capable of converting succinate and lactate into propionate [136,137,138,139]. Succinate and lactate are metabolites abundant in the tumor microenvironment, have pro-inflammatory properties, and participate in the activation of tumor-associated macrophages [140,141]. Lactate is also found increased in IBD patients and correlates with disease severity [142]. *Dialister* abundance is negatively correlated with total carbohydrates and starch ingestion [143] but is increased by the ingestion of dietary fiber [144,145], which would explain the increase in MD. Importantly, a higher anti-inflammatory response driven by the inclusion of whole grains in the diet was observed in subjects with increased proportions of *Dialister* [145]. *Phascolarctobacterium* increased in CRC and decreased in IBD. Its abundance has been associated with lower circulating levels of inflammatory markers [146], and the decrease observed in IBD has already been reported in patients and linked to colon inflammation [147]. An increase in *Phascolarctobacterium* was reported in CRC patients, particularly early-stage patients [148,149], and interestingly, both succinate and lactate are overrepresented in their fecal metabolome [149]. Therefore, its increase in CRC may be related to the augmented succinate and lactate availability and not necessarily to the gut microbiota health or the local inflammation status. The genus *Veillonella*, which was increased in IBD (Figure 4), includes the species *V. parvula,* which expresses a lipopolysaccharide with pro-inflammatory activity [150] and induces strong expression of proinflammatory IL6 in vitro, inhibiting the expression of potentially antitumoral IL12p70 induced by other bacteria [151]. *Veillonella* was also increased and was positively correlated with pro-carcinogenic TMAO levels in IBD patients, and it was further increased in Crohn’s patients with deep ulcers [152,153].

The less frequent *Veillonellaceae* genera *Megasphaera*, *Acidaminococcus*, and *Mitsuokella* are also SCFA producers [154,155,156]. They all use amino acids as a carbon source and have different affinities for them [157]. This observation could explain the different abundances observed, as vegetable sources (soy, rice), which are increased in MD, have less lysine but more histidine than animal sources (whey) [158]. For example, *Megasphaera,* which was low in the MD group, grows well in the presence of lysine but not of histidine [157].

Another taxon of interest is the phylum Verrucomicrobia, mainly composed of the genus *Akkermansia*, which we found increased in MD and decreased in IBD compared to RS, HC, or FC, but also increased in CRC. The presence of this mucolytic bacterium is normally associated with a healthy microbiota [159] and inversely related to inflammation [160]. The increase observed in MD has not been reported before and may be related to the ingestion of fiber, which has been shown to increase mucin expression in animal models [161,162]. The decrease in *Akkermansia* abundance in IBD subjects and the increase in CRC had already been reported [122,148,163]. Its high abundance in cancer is probably associated with changes in mucins expression associated with CRC progression, including an upregulation of MUC5AC and downregulation of MUC2 [164], and even the loss of its expression in some patients [165]. Although a direct association between mucin types expression and mucolytic bacteria abundance has not been demonstrated, an increase of *Akkermansia* has been observed in *Muc2*^-/-^ mice [166], and in the *Winnie* mice strain, which produces an aberrant MUC2 [167]. Conversely, *Akkermansia* is decreased in in vitro culture assays with MUC2 as the sole carbon source, while growth of [*Ruminoccocus*] species is favored [122].

For the phylum Bacteroidetes, we observed a decrease in the *Prevotellaceae* family and its main genus *Prevotella,* in IBD, CA, and CRC compared to HC and MD. The decrease of *Prevotellaceae* was previously reported in UC [168] and CRC patients [148]. Some species have pro-inflammatory properties, while others have anti-inflammatory ones [169,170]. No differences were found for *Bacteroidaceae*, which is another Bacteroidetes family. *Bacteroidaceae* are mainly composed of the genus *Bacteroides* and include both benefic and pathogenic species [171].

The observed differences between diets and patient groups suggest that MD may have a potentially beneficial effect on the patient’s microbiota by increasing beneficial bacteria such as *Akkermansia* and *Adlercreutzia* or the *Ruminococcaceae* family, reducing at the same time potential pathogenic pro-inflammatory bacteria such as *Fusobacterium* or Proteobacteria. Despite the high variability of the data from WD and PD groups, few conclusions could be reached. However, some of the observed effects of WD in microbiota, such as the increases in *Fusobacterium* and *Collinsella,* suggest that this diet may be harmful to patient’s health. For PD, we observed few differences that may be considered beneficial, mainly increases in *Ruminococaceae*, *Coprococcus*, and *Lachnospira*. However, these changes alone could hardly suggest a potential benefit or harm. In all cases, further testing would be needed to confirm their potential therapeutic or harmful effect on patients.

Our study presented some limitations stemming from the limited availability of high-quality 16S public data for certain diets or diseases, which made it difficult to balance the number of subjects in each group, as well as their gender or age. This also led us to choose to not perform analyses below the genus, due to the greater number of unidentified OTUs at the species level compared to all other taxonomic levels. Furthermore, the data used have provided us with valuable information on the bacterial composition but do not allow evaluating their metabolic activity or evaluating other components of the microbiome such as fungi or viruses.

## 5. Conclusions

We conducted a meta-analysis of diets and intestinal diseases related to inflammation and cancer that highlights unique characteristics of the bacterial population associated with MD. The microbiota of subjects following MD was enriched with beneficial bacteria that promote an anti-inflammatory environment, which were instead reduced in IBD, CA, and CRC groups. Conversely, taxa with pro-inflammatory properties that can alter the gut barrier functions were reduced in MD and increased in IBD, CA, and CRC groups. Among the modulated taxa, we reported for the first time an increase in *Akkermansia* and a reduction in *Fusobacterium* in MD, even below the levels observed in healthy subjects without a defined diet. *Akkermansia* is a marker of a healthy gut, and *Fusobacterium* is a known pathogenic bacterium associated with cancer and IBD. Fusobacterium has also a critical role in mediating CRC chemoresistance to oxaliplatin and fluorouracil (5-FU) regimens by activating the autophagy pathway [172]. Our results suggest that MD’s effect on the gut microbiota has the potential to prevent cancer and other inflammation-related diseases of the gut.

## Figures and Tables

**Figure 1 nutrients-13-02234-f001:**
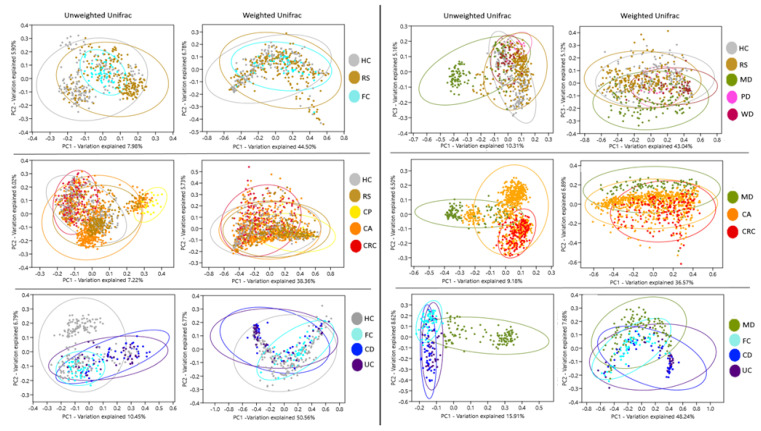
Analysis of bacteria community components. Principal coordinates analysis of Unifrac distances for the different groups. Ellipses indicate 95% confidence intervals. Groups: Healthy subjects (HC), subjects at risk of developing colorectal cancer (RS), healthy familiars of IBD patients (FC). Patients with colon polyps (CP), colon adenoma (CA), colorectal cancer (CRC), Crohn’s disease (CD), and ulcerative colitis (UC). Subjects following a Mediterranean (MD), modern Paleolithic (PD), or Western-like diet (WD).

**Figure 2 nutrients-13-02234-f002:**
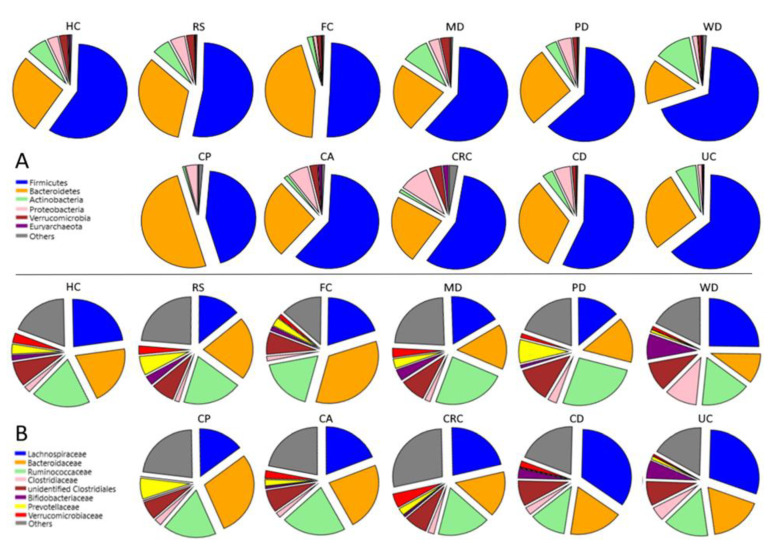
Microbiota composition of the different groups. Relative abundance of the most represented (**A**) phyla and (**B**) families. Groups: Healthy subjects (HC), subjects at risk of developing colorectal cancer (RS), healthy familiars of IBD patients (FC). Subjects following a Mediterranean (MD), modern Paleolithic (PD), or Western-like diet (WD). Patients with colon polyps (CP), colon adenoma (CA), colorectal cancer (CRC), Crohn’s disease (CD), and ulcerative colitis patients (UC).

**Figure 3 nutrients-13-02234-f003:**
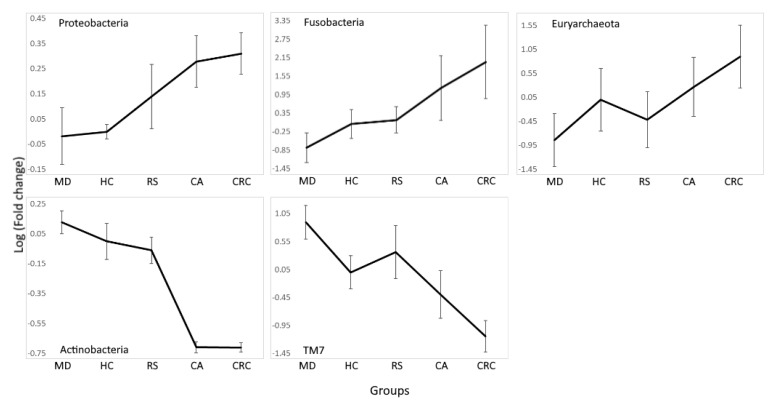
Phyla progressively increased or decreased from subjects in the Mediterranean diet (MD), healthy subjects (HC), subjects at risk of developing colorectal cancer (RS), colon adenoma (CA), and colorectal cancer patients (CRC) groups. Mean log_10_ fold change was calculated using the relative abundance in the HC group as a reference. Significant differences between MD and HC, HC and RS, RS and CA, and CA and CRC (FDR < 0.01) were found for all phyla except for CA and CRC comparison in TM7 (FDR = 0.96) (Appendix A). Euryarchaeota and TM7 presented changes in relative abundance between HC and RS that do not follow the general trend; however, CA was increased in the former phyla and decreased in the latter when compared to both HC and RS.

**Figure 4 nutrients-13-02234-f004:**
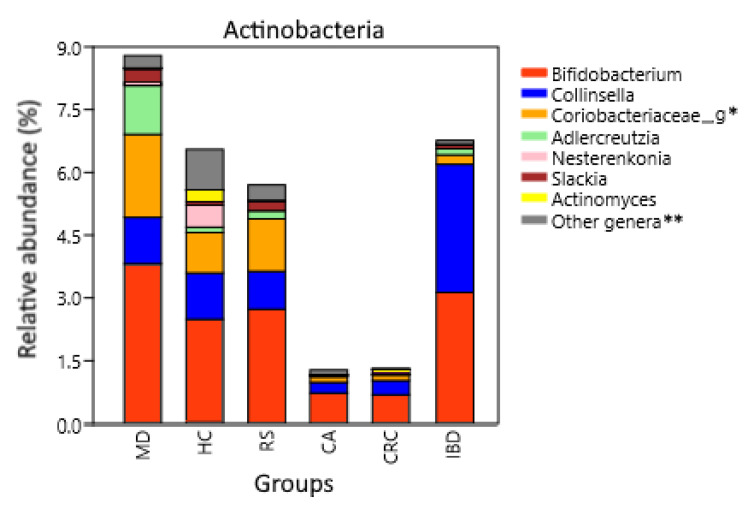
Actinobacteria composition at the genus level. Relative abundance corresponds to the average percentual fraction of reads in the group representing each OTU. Groups: Subjects following a Mediterranean diet (MD), healthy subjects (HC), subjects at risk of developing colorectal cancer (RS). Patients with colon adenoma (CA), colorectal cancer (CRC), or inflammatory bowel diseases (IBD). * Genus or genera unannotated in the reference database. ** Less abundant genera and sequences that could not be classified at the genus level.

**Figure 5 nutrients-13-02234-f005:**
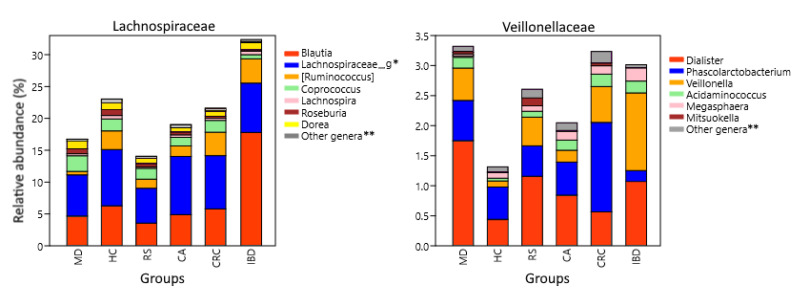
*Lachnospiraceae* and *Veillonellaceae* composition at genus level. Relative abundance corresponds to the average percentual fraction of reads in the group representing each OTU. Groups: Subjects following a Mediterranean diet (MD), healthy subjects (HC), subjects at risk of developing colorectal cancer (RS). Patients with colon adenoma (CA), colorectal cancer (CRC), or inflammatory bowel diseases (IBD). * Genus or genera unannotated in the reference database. ** Less abundant genera and sequences that could not be classified at the genus level.

**Table 1 nutrients-13-02234-t001:** Groups included in the meta-analysis.

Group ^1^	Diet	Disease Status	Subjects	Description	Studies
HC	Undefined	No disease	196	Healthy subjects non-related to patients	[27,47,48,57,58,59]
FC	Undefined	No disease	54	Healthy familiars of IBD patients	[60]
RS	Undefined	No disease	214	Subjects at risk of CRC ^2^	[44,57,61,62,63]
WD	Western-type	No disease	38	Diet subjects. Healthy	[42]
PD	Modern Paleolithic	No disease	15	Diet subjects. Healthy	[43]
MD	Mediterranean	No disease	123	Diet subjects. Healthy	[61,62,64]
CP	Undefined	Colon polyposis	23	Patients	[63]
CA	Undefined	Colorectal adenoma	662	Patients	[57,63,65,66,67]
CRC	Undefined	Colorectal cancer	155	Patients	[57,65,66]
UC	Undefined	Ulcerative colitis	38	Patients	[59,60]
CD	Undefined	Crohn’s disease	45	Patients	[58,59,60]

^1^ Groups: Healthy subjects (HC) and healthy familiars of IBD patients (FC). Subjects at risk of developing colorectal cancer (RS), patients with colon polyps (CP), colon adenoma (CA), colorectal cancer (CRC), Crohn’s disease (CD), and ulcerative colitis patients (UC). Subjects following a Mediterranean (MD), modern Paleolithic (PD) or Western-like diet (WD). ^2^ Further criteria information in Appendix A.

**Table 2 nutrients-13-02234-t002:** Average OTU composition (%) of the different groups at phylum level.

Phylum ^1^	HC	RS	FC	MD	PD	WD	CP	CA	CRC	IBD
Firmicutes	58.9506	52.9856	50.5467	60.7094	62.6186	68.4219	43.9497	60.7098	57.1596	57.2953
Bacteroidetes	27.2735	33.3556	44.5428	23.4625	27.3412	15.0817	50.2231	27.0028	23.5202	30.5508
Proteobacteria	3.4721	4.7896	1.1461	3.3301	4.4188	1.6484	3.8428	6.5936	9.7906	3.8073
Actinobacteria	6.5486	5.6996	1.7042	8.7902	3.6509	11.9093	0.6523	1.2800	1.1112	6.7657
Verrucomicrobia	2.6702	2.5748	1.2910	3.1041	1.4947	0.9764	0.0091	2.5214	3.9575	0.8956
Euryarchaeota	0.5848	0.2252	0.2990	0.0847	0.0536	0.4117	0	1.0877	1.7885	0.0168
Fusobacteria	0.0146	0.0197	0.0029	0.0025	0.0042	0.0848	0.0117	0.2145	2.0454	0.2526
Tenericutes	0.3459	0.1043	0.3512	0.0482	0.2094	0.7785	0.0971	0.3692	0.2732	0.0329
Cyanobacteria	0.0581	0.1052	0.1002	0.2465	0.1488	0.2541	0.0536	0.0625	0.0579	0.0284
Synergistetes	0.0098	0.0513	0.0014	0.0441	0.007	0.0041	0.4129	0.0848	0.1576	0.0005
Chloroflexi	0.0024	0.0342	0.0008	0.0047	0	0.0179	0.4694	0.03	0.0011	0.0007
TM7	0.0083	0.0191	0.0055	0.0653	0.007	0.0039	0.0013	0.0033	0.0014	0.0210

Groups: Healthy subjects (HC), subjects at risk of developing colorectal cancer (RS), and healthy familiars of IBD patients (FC), Subjects following a Mediterranean (MD), modern Paleolithic (PD) or Western-like diet (WD). Patients with colon polyps (CP), colon adenoma (CA), colorectal cancer (CRC), and inflammatory bowel diseases (IBD). ^1^ Phyla with an average relative abundance > 0.01%.

**Table 3 nutrients-13-02234-t003:** Average OTU composition (%) of the different groups at family level.

Family ^1^	HC	RS	FC	MD	PD	WD	CP	CA	CRC	IBD
*Bacteroidaceae*	20.0833	21.4943	34.1572	15.4143	15.5215	10.1769	28.9815	23.0839	15.5119	17.7059
*Ruminococcaceae*	19.1884	19.3032	17.7934	23.7961	26.1617	15.8293	17.3777	21.0300	17.0954	12.9261
*Lachnospiraceae*	23.0466	13.8709	19.8088	16.7415	13.7672	25.2004	14.8706	19.0354	21.5969	32.3826
*Prevotellaceae*	3.4126	6.4874	2.9435	3.3205	7.9948	0.9122	7.1719	2.1619	2.4454	0.5454
*Verrucomicrobiaceae*	2.5702	2.5748	1.3390	3.1041	1.4059	1.0218	0.0091	2.6860	4.7985	1.4723
*Enterobacteriaceae*	1.8800	2.3051	0.1756	1.4505	1.5145	0.9643	2.8031	3.1807	4.6496	2.8912
*Rikenellaceae*	1.2078	2.3913	3.3995	0.9956	0.6892	1.3703	1.5607	3.7645	1.3509	1.4844
*Veillonellaceae*	1.3149	2.6069	1.5675	3.3201	3.1355	0.9408	1.4446	2.0480	3.2348	3.0156
*Erysipelotrichaceae*	1.3873	2.9033	1.6074	3.5306	2.7406	3.0238	2.0810	2.1707	1.2688	1.9607
*Clostridiaceae*	2.1895	1.5247	1.5840	1.8002	2.9558	10.5602	2.4377	1.8124	2.0468	3.6841
*Bifidobacteriaceae*	2.4882	3.0881	1.4775	3.8193	0.9905	8.5908	0.1076	0.7255	0.6851	4.1287
*Porphyromonadaceae*	1.4686	1.9259	2.1814	1.3922	0.6344	0.5258	1.9845	2.1017	2.6539	0.7964
*Coriobacteriaceae*	2.2984	2.8772	0.1868	4.8781	2.4784	3.3779	0.3076	0.4466	0.5358	3.5795
*Streptococcaceae*	0.7604	1.1587	0.1391	1.1514	1.6407	1.5708	0.8787	0.8000	1.3647	2.4764
*[Barnesiellaceae]*	0.8793	1.0808	0.7642	0.4972	0.5040	0.4677	0.4404	0.8730	0.4621	0.1233
*[Paraprevotellaceae]*	0.4943	0.8159	0.3284	0.8542	0.8998	0.6662	3.1720	0.7737	0.6244	0.0257
*S24–7*	0.6452	0.8367	0.4641	0.5315	0.2762	0.3106	0.6800	0.5968	0.9590	0.1841
*Methanobacteriaceae*	0.6146	0.2509	0.2882	0.0847	0.0512	0.4403	0	0.6862	1.5798	0.0292
*Alcaligenaceae*	0.6979	0.8799	0.5232	0.2415	0.6768	0.2323	0.1827	0.2702	0.2863	0.3770
*[Odoribacteraceae]*	0.5183	0.2700	0.1939	0.2048	0.3140	0.2704	0.1536	0.2848	0.3808	0.0449
*Desulfovibrionaceae*	0.2213	0.4774	0.1547	0.2684	0.2958	0.1048	0.1304	0.2288	0.3753	0.1004
*Peptostreptococcaceae*	0.2905	0.3369	0.0330	0.3157	0.2815	0.2195	0.5588	0.1831	0.4749	0.5393
*Christensenellaceae*	0.2410	0.1657	0.2954	0.1624	0.5880	0.0704	0.1304	0.2189	0.7917	0.0163
*Sphingomonadaceae*	0.0051	0.0001	0.0001	0.0007	0	0.1645	0.0001	0.3026	1.3384	0.0001
*Lactobacillaceae*	0.2327	0.0474	0.0032	0.3848	0.0170	0.1321	0.3623	0.1945	0.3568	0.4582
*Fusobacteriaceae*	0.0173	0.0191	0.0035	0.0020	0.0040	0.0519	0.0030	0.2023	1.1520	0.2526

Groups: Healthy subjects (HC), subjects at risk of developing colorectal cancer (RS), and healthy familiars of IBD patients (FC), Subjects following a Mediterranean (MD), modern Paleolithic (PD), or Western-like diet (WD). Patients with colon polyps (CP), colon adenoma (CA), colorectal cancer (CRC), and inflammatory bowel diseases (IBD). ^1^ Families with an average relative abundance > 0.01%.

## Data Availability

The data presented in this study are openly available in the NCBI short reads archive (SRA), accessions PRJEB2165 [45], PRJNA318004 [55], PRJNA388210 [56], PRJNA450340 [57], PRJNA324147 [58], PRJEB39062 [59], PRJNA510080 [60], PRJNA534511 [61], PRJEB14782 [40], PRJNA247489 [62], PRJEB6070 [64], PRJNA516932 [65]; or in MG-RAST, project IDs mgp6248 [27], mgp80356 [46], mgp88216 [44], mgp89161 [41]; or in the Microbiome CRC Biomarker Study data file repository [63,173].

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
