# Peer review of "Mediterranean Diet to Prevent the Development of Colon Diseases: A Meta-Analysis of Gut Microbiota Studies"

_nutrients, 2021, doi:10.3390/nu13072234_

Round 1
Reviewer 1 Report
This is a joint meta-analysis of publicly available 16S data derived from 15 subjects following MD or other diets and from patients with CRC, IBD or other gut-related diseases 16 was conducted. The methods in this manuscript are appropriate and well researched in detail.
1.At this point, it has not been determined that MD can reliably prevent CRC or IBD, so I am concerned that the way this manuscript is written will mislead people into thinking that the preventive effect of MD has been determined. I would like to see the entire manuscript worded in such a way that MD has the potential to prevent CRC and IBD.
- References you have used to evaluate the benefits of MD include insights and literature on the potential effects of MD. Please use references that it was definitely effective, or you can state that it is not yet certain, but the possibility has been suggested.
Introduction
- Lin-.40-41: Benefits of MD have been demonstrated on several pathologies, including cancer [10-13].
These references do not conclude that MD itself is effective. Please use references that it was definitely effective. Otherwise, state that the possibility has been suggested.
Excerpt from References
10:It is observed that the adoption of Mediterranean or DASH-type dietary patterns may contribute to the prevention of HF, but these results need to be analyzed with caution due to the low quality of evidence.
12: assessed in clinical trials, specific micronutrients showed a limited benefit. Further research is required to evaluate the role of individual food compounds and complex nutritional interventions with the potential to decrease inflammation as a means of prevention and management of IBD.
13: Olive oil polyphenols, red wine resveratrol, and tomato lycopene showed several characteristics in vitro that interfere with molecular cancer pathways. At the same time, many clinical studies have reported an association of these components with a reduction in cancer initiation and progression.→not MD
2.Line51-53
Contrary to this normal state, the term dysbiosis refers to an imbalance in the microbiota composition and metabolism, which leads to proinflammatory conditions and contributes to the pathogenesis of various diseases, ranging from psychiatric diseases to metabolic disorders and cancer [18,20,21].
→The cited references do not allow us to say this much. Please rephrase it as suggesting that it may be an etiological factor in a variety of diseases.
- Lin-.64-71: In this section, you will write about the aims of the research and the policies to achieve the aims of the research. However, you are writing a summary of your results here. Please rewrite it.
Discussion
Even for meta-analyses, there are limitations to the manuscript.
Reviewer 2 Report
In the current study, the authors conducted a joint meta-analysis of publicly available 16S data derived from subjects following different diets and from patients with CRC, IBD or other gut-related diseases. Their findings are interesting, but there are numerous details missing and therefore it requires extensive re-interpretation and additional details/experiments before it can be properly evaluated.
(1) What is the association between PD and microbiota? The authors should provide some background information about this diet.
(2) What about the data from patients with MD or WD? Will these diets alleviate or aggravate the disease and change the trend of gut microbiota?
(3) Table 1: The authors should provide more information about the study subjects, such as age and sex. These factors also affect the composition of gut microbiota.
(4) Figure 1: What is the difference between weighted and unweighted Unifrac? It looks like unweighted unifrac models had more clear separation than weighted ones. The authors should give some explanations.
(5) Tables 2-3, Figure 3: The authors should provide standard deviation.
(6) Figures 1-5: The authors should provide abbreviation information.
(7) Figures 4-5: The authors should consider combine these two figures.
Round 2
Reviewer 2 Report
The manuscript has been significantly improved and now warrants publication in Nutrients.